# Effects of an App-Based Intervention to Improve Awareness and Usage of Early Childhood Intervention Services During the COVID-19 Pandemic: Randomized Controlled Trial of the CoronabaBY Study from Germany

**DOI:** 10.3390/healthcare13162000

**Published:** 2025-08-14

**Authors:** Catherine Buechel, Volker Mall, Ina Nehring, Anna Friedmann

**Affiliations:** 1Social Pediatrics, School of Medicine and Health, Technical University of Munich, 81377 Munich, Germany; volker.mall@tum.de (V.M.); ina.nehring@tum.de (I.N.); anna.friedmann@tum.de (A.F.); 2German Center for Child and Adolescent Health (DZKJ), Partner Site Munich, 80337 Munich, Germany

**Keywords:** early childhood intervention, awareness, usage, COVID-19 pandemic, psychosocial stress, randomized controlled trial, app-based intervention

## Abstract

**Background/Objectives**: Young families experiencing high levels of psychosocial stress should be addressed at an early stage to counteract potential negative effects on the parents’ and child’s wellbeing. This became particularly relevant during the COVID-19 pandemic when especially families have been strained by restriction measures. Early Childhood Intervention services (ECI) can provide low-threshold support, to which access should be encouraged. **Methods**: The randomized controlled trial of the CoronabaBY study with N = 1384 participants focuses on the effects of a newly developed app-based intervention on the awareness and usage of ECI services by young families with infants and toddlers during the COVID-19 pandemic in Germany. The analysis compares knowledge and usage rates of an intervention group (IG) and a waitlist control group (WCG) at three survey stages (pre-test, post-test and follow-up). **Results**: There was a significant increase in awareness of ECI services from pre-test to post-test in both the IG (15.2%) and WCG (10.7%) (*p* < 0.001), but the difference between the two groups was not significant. In contrast, the usage rate of further ECI services at post-test was significantly higher in the IG (12.1%) compared to the WCG (8.2%) (V = 0.060, *p* = 0.012), especially in a subgroup of highly stressed parents. **Conclusions**: Digital applications show potential to improve access to ECI services and should be continuously developed and evaluated to appropriately address young families and their needs.

## 1. Introduction

Families facing challenging living conditions may develop psychosocial stress and an increased need for support [1,2,3,4]. If stressors persist, this can negatively affect mental health in the long term [5,6,7]. Potential risk factors are demands in the care of the child [8,9,10], sociodemographic factors (e.g., poverty, [1,11], familial and interpersonal (e.g., conflicts in partnership) as well as biographical factors (e.g., critical life events/childhood experiences) [2].

Additionally, broader social crises have the potential to increase psychosocial stress, as was the case with the COVID-19 pandemic [12]. Families have been particularly strained by restriction measures which have led to disruptions in everyday routines [13], additional childcare responsibilities [14] and a potential increase in social isolation [15], due to limited access to family support services [13] and daycare facilities [14], and the widespread obligation to work from home. Psychosocial stress factors and symptoms in families thus became noticeable in the course of time, showing up as increased parenting stress [16,17,18,19,20] and significant depression and anxiety symptoms in parents [3,21,22]. Parenting stress arises when resources in child care and the demands of parenthood are out of balance [8,10]. This can affect parent–child relationships, which were actually increasingly strained during the pandemic [23,24], and may result in adverse development in the child [25,26]. Due to its high practical significance, parenting stress is often considered an indicator that shows a need for support in young families. Higher levels of psychological problems were also observed among school-age children and adolescents [27,28,29].

The vulnerable group of families with infants and toddlers has been paid less attention than other age groups in research on the psychosocial consequences of the pandemic, even though early-life stress is a major risk factor for mental health across the life span [30,31,32]. In Germany, there is a system of voluntary, cost-free Early Childhood Intervention services (ECI), organized in municipal networks which address families from pregnancy onwards through the first three years of a child’s life. ECI services offer low-threshold support tailored precisely to the needs in early childhood and challenging life situations [33].

Support and education of parents of this very young target group is particularly relevant, as infants and toddlers are most dependent on their caregivers [34]. It has been proven that parenting interventions for young children enhance parental competencies in care [35], promote the development of a strong and sensitive relationship with the child [33], and effectively improve early childhood development with positive benefits on child cognitive, socioemotional, language and motor development [35]. By these means, threats to the child’s well-being can be prevented and risks to development can be identified at an early stage [36].

Especially in times of crisis, such as the COVID-19 pandemic, a comprehensive knowledge of and adequate access to ECI services should be considered important against the background of increased psychosocial stress factors and support needs [16,37]. However, access to all family services from childhood to adolescence was limited and temporarily shifted to digital or telephone contact [13]. Accordingly, a digitization boost could also be seen for ECI [38], although these offers are predominantly analogue with a focus on personal contact [39]. Even if a certain degree of accessibility to services was ensured this way, the number of contacts still fell massively at the beginning of the pandemic compared to before [13]. Access to ECI services overall was limited. Usage rates fell by almost 30%, although 40% of the parents would have used more offers if it had been possible [40]. Due to the far-reaching consequences of the COVID-19 pandemic, it was of great importance to ensure awareness of and access to ECI services, particularly for the severely affected families showing high parenting stress. In fact, there is potential assumed to optimize accessibility to ECI services for families in stressful situations [41].

### Study Aim

To our knowledge, there is limited research addressing the improvement of awareness and usage of Early Childhood Intervention services (ECI) in the target group of families with children up to 3 years old during the pandemic. The CoronabaBY study focused on these topics in a randomized controlled trial (RCT).

The presented RCT addresses the research questions of how a newly developed app-based intervention (i.e., a digital information module on ECI) affected awareness and usage of ECI services in young families during the COVID-19 pandemic. We made the following hypotheses:Parents who accessed the app module (intervention group, IG) would have a significantly higher awareness of ECI at post-test compared to parents who did not access the app module (waitlist control group, WCG).Parents in the IG would have a significantly higher usage rate of ECI at post-test compared to parents of the WCG.These effects in awareness and usage of ECI between IG and WCG would also show in the follow-up.

We furthermore aimed to know whether the awareness and usage of ECI services varied depending on the severity of parenting stress. We made the following hypotheses:The more stressed the parents were, the higher the awareness of ECI was—both for the pre-test, the post-test and the follow-up.The more stressed the parents were, the higher the usage rate of ECI was—both for the pre-test, the post-test and the follow-up.

In addition, we aimed to analyze the effectiveness of using the app module on the knowledge and usage of ECI services in the subgroup of highly stressed parents. According to the parenting stress index (PSI), high scores indicate an increased need for support due to a lack of parental resources to provide adequate care for and to maintain an appropriate relationship with the child [8]. We hypothesized the following:Highly stressed parents who accessed the app module (intervention group, IG) would have a significantly higher awareness of ECI at post-test and follow-up compared to highly stressed parents who did not access the app module (waitlist control group, WCG).Highly stressed parents of the IG would have a significantly higher usage rate of ECI at post-test and follow-up compared to highly stressed parents of the WCG.

## 2. Materials and Methods

### 2.1. Study Design

The CoronabaBY study focuses on psychosocial stress during the COVID-19 pandemic (‘Corona’) in families with infants and toddlers (‘baby’) from Bavaria, Southern Germany (‘BY’), and comprises several study arms (for further details see [37]). This paper reports on a randomized controlled trial (RCT), that tested the effects of using a newly developed app info module during the COVID-19 pandemic on awareness and usage of Early Childhood Intervention services by comparing the outcomes of an intervention group and a waitlist control group. CoronabaBY is a single-center, cluster-randomized study with multiple recruitment sites (clusters = participating pediatric practices). The study center is the Chair of Social Pediatrics at the Technical University of Munich.

Data were collected continuously from the 1st of February 2021 until the 2nd of November 2022.

The study protocol was approved by the Ethics Committee of the Technical University of Munich (vote no. 322/20 S) on 15 June 2020, and pre-registered in OSF (https://osf.io/search/?q=tksh5&page=1, accessed on 9 August 2025).

### 2.2. Recruitment and Procedure

Parents with children up to three years were recruited and surveyed via the smartphone app ‘Meine pädiatrische Praxis’ (‘My pediatrician’) (www.monks-aerzte-im-netz.de, accessed on 9 August 2025), which is a well-established communication tool connecting parents with their pediatrician. Recruitment took place within the patient base of participating pediatric practices. All pediatricians in Bavaria (Southern Germany) using ‘My pediatrician’ as part of their practice management were invited to participate in the study (N = 300). After giving informed consent (N = 73, response rate = 24.3%), the participating pediatricians were cluster-randomized and allocated to intervention group practices (IG) and waitlist-control group practices (WCG) by the app programmers via a computer-generated random allocation sequence. Families who were patients in an intervention practice were automatically participants of the intervention group, families who were patients in a waitlist control practice were, accordingly, participants of the waitlist control group. Subsequently, a study invitation (including detailed information and informed consent form) was sent out together with an invitation to the next early childhood checkup (‘U-Untersuchung’) to all eligible parents in the app, i.e., having a child up to 3 years of age, using the app, understanding German. Further selection of participants was not possible due to the pseudonymized conduct of the study within the app, in which certain information (e.g., on psychiatric diagnoses) was not provided. Informed consent was given via the app.

Measurement time depended on time of early childhood checkup. At pre-test, the first checkup considered was ‘U4’ (child aged 3–4 months) and the last was ‘U7a’ (child aged 34–36 months), thus the ages of the children ranged from 3 months to 3 years. Post-test took place 3 months after the pre-test, the time of follow-up was linked to the subsequent checkup after the post-test. Corresponding reminders via app acted as invitations to the in-app-pre-test, post-test and follow-up surveys (for detailed recruitment information, see also [37]). Families who were patients of IG practices received access to a newly developed information module on Early Childhood Intervention services within the app after the pre-test questionnaire was completed. The WCG practices’ patient families only received access to the information module after having completed the follow-up questionnaire.

Participants were not blinded to the study conditions.

### 2.3. Intervention

The aim of the information module, designed as an additional offer within the app ‘My pediatrician’ for parents of children up to 3, was to provide information about Early Childhood Intervention services (ECI) and supporting access to it. The module could be accessed in a special service section of the app, to which the intervention group was immediately directed after pre-test. It was designed to be low-threshold and compact with the aim of quickly providing relevant information about ECI support options and linking users to specialized websites, registers and search masks. From the app module’s starting pages (see Appendix A), parents could obtain access to various briefly presented topics via corresponding buttons (see Figure 1): information about ECI and the Early Childhood Network in Bavaria (‘KoKi’), self-care tips (‘What you can do for yourself’), information providing ‘first aid for difficult situations’, e-mail and phone numbers in case of ‘acute emergencies’ (e.g., violence), links to address directories and maps of advice centers in Bavaria (‘parenting counseling centers’, ‘cry baby counseling’) as well as links to websites with further information on children‘s health, on support options in case of problems and overloads, and with tips for parents in times of Corona (‘helpful links’). The module was presented in German. Parents having access could use the module within the app unlimitedly and independently.

### 2.4. Measures

All data were collected by standardized questionnaires via the app. All participants were asked about general sociodemographic characteristics at pre-test (t1); the IG also answered questions about usage and impression of the app information module at post-test (t2). Knowledge and usage of Early Childhood Intervention services were covered at pre-test, post-test and follow-up. Parenting stress was assessed at pre-test and follow-up.

#### 2.4.1. Usage and Evaluation of the App-Based Intervention

The intervention group was asked questions about the usage of the app information module at the post-test: whether they had accessed (yes/no) and actively used (yes/no) it, what their overall impression was (answers given on a German school grading scale from 1 ‘very good’ to 6 ‘insufficient’) and which information they had found helpful (answers given on a 3-point scale regarding the module topics: very helpful, not very helpful, not helpful at all).

#### 2.4.2. Awareness and Usage of Early Childhood Intervention

Ten questions were asked about knowledge and usage of Early Childhood Intervention services for families with children up to 3 years of age in Germany: whether the parents knew such services exist (yes/no) (‘When you have a baby or toddler, it sometimes feels like things are getting out of hand. Do you know that there are free, non-binding support services for young families for such situations (so-called ‘Frühe Hilfen’—this includes, for example, parenting advice, crybaby advice, baby/child groups)?’), and if so, how they learned about them (list with multiple answer options, e.g., pediatrician, gynecologist, family/friends), if they already had made use of them (yes/no) and how they rated accessibility (4-point-scale regarding waiting time: none, few days, few weeks, several weeks).

#### 2.4.3. Parenting Stress

To assess parenting stress, the parent domain of the German Version of the ‘Parenting Stress Index (PSI)’ (‘Eltern-Belastungs-Inventar’ EBI; [8]) was applied. High scores indicated limited parental resources for upbringing and care for the child. The parent domain includes 7 subscales assessing impairments in parental functioning: attachment, social isolation, doubts about parental competence, depression, health, role restriction and spouse-related stress.

Answers were given on a 5-point Likert scale ranging from 1 = strongly agree to 5 = strongly disagree, resulting in a possible score range of 28 to 140. The three cut-off categories for each subscale and the whole parent domain were ‘not stressed’ (T-value < 60), ‘stressed’ (T-value = 60–69), and ‘strongly stressed’ (T-value ≥ 70) [8]. Accordingly, a high score indicates a high level of parenting stress. For the present analyses, only the total score of the parent domain was applied.

Internal consistency of the parent domain has been proven to be good (α = 0.93), and retest reliability after one year is r = 0.87. Correlations with stress indicators and related constructs resulted in the assumption of test validity [8,42].

### 2.5. Statistical Analysis

#### 2.5.1. Power

An a priori calculation of the sample size was performed using G*Power software 3.1. A comparison between two groups (IG and WCG) was assumed. For the analysis of differences, it was planned to calculate frequencies and use Chi-square-tests. The estimation was based on an α = 0.05 and a power of 1 − β = 0.80. Due to the lack of appropriate reference studies, small effects were assumed (ω = 0.1). The calculation resulted in a sample size of 1283 participants.

#### 2.5.2. Analysis Plan

Primary Outcome

We tested how the newly developed app-based intervention affected awareness and usage of Early Childhood Intervention services (ECI) in young families during the COVID-19 pandemic by comparing these outcomes between IG and WCG at pre-test, post-test and follow-up. To ensure that effects at post-test and follow-up can be attributed to the intervention (i.e., the app information module), only those participants of the IG who had at least accessed the module were included in the post-test and follow-up analyses. In a second step, we carried out an intention-to-treat (ITT) analysis, to complete the results and draw more precise conclusions. This included all participants in the IG, also those who had not accessed the app information module.

To counteract potential bias in the analysis of awareness of ECI services, we excluded all participants who had stated in the pre-test that they had previously used ECI services. We did not apply this filter to the usage analysis of ECI services because we wanted to evaluate all usage (not just initial usage).

Usage of the app module as well as awareness and usage of ECI services were calculated in frequencies. Differences were analyzed via Chi-square-tests due to the nominal scale levels; effect sizes were calculated by Cramer’s V.

Secondary Outcome

In a sub-analysis, furthermore, we tested whether the awareness and usage of Early Childhood Intervention (ECI) varied depending on the severity of parenting stress according to the three EBI cut-off categories: not stressed, stressed, strongly stressed. In addition, we analyzed the effectiveness of the app-module on awareness and usage of ECI in the subgroup of highly stressed parents by comparing those outcomes between IG and WCG. Differences were analyzed for pre-test, post-test and follow-up by calculating frequencies, Chi-square-tests and Cramer’s V.

All the results described were based on an alpha level of 5%. Analyses were performed in IBM SPSS Statistics Version 29.0.

## 3. Results

### 3.1. Participant Enrollment and Sample Characteristics

From 300 pediatricians using the app ‘My pediatrician’, 73 agreed to participate in the study. Within the patient base of these practices, approx. 17,800 eligible parents were invited to the study (figures extrapolated based on available case numbers from the first cross-sectional survey of the CoronabaBY study, see [37]), nearly 4000 gave informed consent and 3305 finally completed the pre-test (t1). Among them, 2002 were allocated to the IG and 1303 to the WCG according to the randomization of their pediatric practice. This difference in case numbers is due to the varying sizes of the practices and occurred randomly. At post-test (t2), 1309 questionnaires were filled out by the IG (of those, 488 parents had at least accessed the app-based intervention) and 896 questionnaires were finished by the WCG. At follow-up, data of 284 parents of the IG could be analyzed (i.e., parents having accessed the intervention) as well as data of 488 parents of the WCG. Figure 2 shows the CONSORT participant flow diagram.

Sample characteristics were collected at pre-test (t1). Most of the participants were mothers (92.9%) with mother tongue German (92.3%) and a mean age of 33.7 years (*SD*: 5.0). Children were on average 16.7 months old (*SD*: 12.1). IG and WCG did not differ significantly except for financial status (see Table 1).

### 3.2. App-Based Intervention—Usage and Evaluation

In the post-test survey (t2), participants of the IG were asked if they had made use of the newly developed app info module on Early Childhood Intervention services (ECI) and how they rated it. Around a third (37.3%) said they had at least accessed the module and one fifth (20.6%) had used it actively (i.e., read information, followed links, used phone numbers, etc.).

The overall impression of the app info module was good to very good. On a scale from 1 (very good) to 6 (insufficient), 20.4% gave it a very good rating, 55.1% rated it as good (i.e., ‘2’), 17.6% as satisfactory (i.e., ‘3’) and 5% as sufficient (i.e., ‘4’). Just under 2% rated the module as poor (1%) or insufficient (0.8%) (i.e., ‘5’and ‘6’). The following topics have been reported as very helpful by the users: information for acute emergencies (72.45%), help for difficult situations (59.8%) and contact links to ECI services (57%).

### 3.3. Awareness and Usage of Early Childhood Intervention Services According to Study Group

#### 3.3.1. Awareness According to Study Group

At pre-test (t1), almost three-quarters of the intervention group (IG) (70.6%) and the waitlist control group (WCG) (73.2%) knew of Early Childhood Intervention services (ECI), with no significant difference between the two groups. When comparing participants in the IG who accessed the app module with the WCG at post-test and follow-up with respect to awareness of ECI services, no significant differences could be detected either (see Table 2). In both groups, awareness was significantly higher at post-test compared to pre-test. This also applied after conducting the intention-to-treat (ITT) analysis.

When asked how parents knew about the offers, they most often said posters/flyers, followed by pediatrician and family/friends (btw. 19% and 32%, see Figure 3). In the follow-up questionnaires, around 20% also cited the app as a source of knowledge.

#### 3.3.2. Usage According to Study Group

When asked at pre-test (t1) whether they had used Early Childhood Intervention services (ECI) before, around a third of all participants (35.6%) answered yes. No significant difference was found by the study group (see Table 2). When asked at post-test (t2) if they had used further ECI services since the last survey stage, significantly more parents of the IG (12.1%) than of the WCG (8.2%) affirmed it (*p* = 0.012). Almost 10% of the parents in each group answered the same question in the follow-up with yes (no significant difference) (see Table 2). The intention-to-treat analysis (ITT) revealed a lower proportion of parents in the IG (6.0%) who had used further ECI services since the last survey stage until post-test, compared to 8.2% in the WCG (*p* = 0.014).

Most parents initiated the contact to the support services themselves (89.1%), 6.1% were referred by the pediatrician, 4.6% by the gynecologist and 4.3% by the KoKi. Of the parents who made use of ECI services, most had immediate access (36.1%) or waited only a few days (47.8%).

### 3.4. Awareness and Usage of Early Childhood Intervention Services According to Parenting Stress Levels

#### 3.4.1. Awareness According to Parenting Stress Levels

When comparing subgroups of parents according to different levels of parenting stress, the parents in the highly stressed group knew least about Early Childhood Intervention services (ECI) (51.5%), while non-stressed parents most frequently knew of these services (76.6%) (see Table 3). This was true for pre-test and follow-up. In all subgroups, the proportion of parents who knew of ECI significantly increased from pre-test to follow-up.

When considering the highly stressed parents by study group, assuming this population has the greatest need for support, there is no significant difference regarding awareness of ECI between IG and WCG, neither at the pre-test, nor at the post-test or follow-up.

#### 3.4.2. Usage According to Parenting Stress Levels

When looking at the usage of Early Childhood Intervention services (ECI) according to parenting stress levels, the usage rate was highest for the highly stressed parents (47.2%), followed by the stressed parents (38.9%) and the non-stressed parents (31.2%) (see Table 3). This was true for pre-test (ever used services) as well as for the follow-up (further usage of services since last survey stage).

When considering the highly stressed parents by study group, there is a significant difference regarding usage of ECI services between IG and WCG at the post-test. Nearly one third (31.4%) of the highly stressed parents in the IG stated they had used a further ECI service since the last survey stage while only 15.4% said so in the WCG (*p* = 0.014, Cramer's V = 0.177). At the pre-test and the follow-up, there were no significant differences regarding usage of ECI services between IG and WCG in the group of highly stressed parents.

## 4. Discussion

The presented randomized controlled trial showed that the proportion of parents who knew of Early Childhood Intervention services (ECI) increased overall from nearly three-quarters at pre-test to almost 90% at post-test, but there were no significant differences between intervention group (IG) and waitlist control group (WCG). Usage rates were lower than awareness rates (around a third of all parents had ever used ECI), but a significantly higher proportion of the IG parents who had at least accessed the intervention module had used a further ECI service by the post-test stage, compared to the WCG. This is also evident when only considering the highly stressed parents in each group. Overall, the usage rates were highest for the highly stressed parents compared to stressed and non-stressed parents, whereas awareness of ECI services was highest for the non-stressed parents and lowest for the highly stressed. This was true for all survey stages.

Looking at the results in more depth, ECI services were known to most of the parents. The awareness rates at pre-test (IG: 70.6%, WCG: 73.2%) were comparable to the average knowledge of the various ECI services before the pandemic [43]. Data also show that the level of knowledge can be raised even higher: while 70.6% of the IG and 73.2% of the WCG already knew those support services at pre-test, 90.6% of the IG-parents and 86.1% of the WCG-parents said they knew about them at follow-up. More than a fifth of all parents stated in the follow-up that they knew ECI services through the app. In between, the parents had either accessed the app info module where those services were mentioned in detail (IG), or they were made aware of the topic through the in-app questionnaires about ECI at pre-test (WCG). Since there was an increase in awareness in both groups, but without a significant difference, the effect of the app module cannot be determined, but cannot be completely ruled out either. At this point, other influencing factors that could not be controlled for may also have been relevant. Study participation itself may have had an impact by asking parents about their knowledge and usage of ECI. The fact that the study was placed and conducted within the app and there was an overall improvement in knowledge of ECI services nevertheless suggests the potential of digital applications to communicate ECI in pediatric practice. The extent to which the app-based module, the app itself, or study participation had an impact would have to be re-examined in a more strictly controlled setting (e.g., by including a group with no app usage).

When looking at usage of ECI services more closely, around a third of all participants (35.6%) answered at pre-test they had already used such services. German comparative studies on ECI services reveal usage rates for each service individually, which vary between approximately 5% (services for special needs/disabilities) and almost 90% (universal preventive services such as midwifery assistance) [40,44]. Considered by levels of parenting stress, the usage rates of ECI services were highest for the highly stressed parents (47.2%), followed by the stressed parents (38.9%) and the non-stressed parents (31.2%). This result reflects the increased need for support in families facing challenging conditions, such as multiple stress factors, high levels of stress and/or mental illnesses [4,33]. When looking at the effect of the app-based intervention on usage rates, significantly more IG parents (12.1%) who had accessed the intervention module than WCG parents (8.2%) affirmed at post-test to have used further ECI services since pre-test (*p* = 0.012). This difference becomes even more evident when looking only at the highly stressed parents, where 31.4% in the IG said that they had used further ECI services compared to 15.4% in the WCG (*p* = 0.014). Among parents in the IG who accessed the intervention module, higher usage rates were evident at the post-test compared to the WCG. The assumption of an intervention effect is supported by the fact that the direction of the difference between the IG and the WCG changes in the ITT analysis when all participants in the IG (including those who did not access the app module) were included. Here, the usage rate of further ECI services at post-test is even lower in the IG than in the WCG (6% compared to 8.2%). Although the results indicate a benefit of the digital intervention as a low-threshold channel to increase the usage rate of ECI services in families in need of support, no causal conclusions can be drawn. We assume that the module usage contributed along with other factors (e.g., measurement-induced changes) for which we could not control. What exactly led to a significantly higher usage rate of further ECI services at post-test among parents in the IG who had accessed the intervention module cannot be conclusively stated, especially regarding comparable increases in awareness in the IG and the WCG. Understanding the module as a mediation channel could be supported by the fact that the module pages on acute situations, which provide contact links, phone numbers and helpful tips, were rated as most helpful by IG-parents. Given the module’s low usage rates in the IG, however, the conclusions should be drawn with caution, especially regarding external validity. Nevertheless, the present RCT was the first intervention study ever conducted in the app ‘My pediatrician’, and parents today might be more likely to use such an app information module.

Maintaining and increasing accessibility to ECI services should remain a priority, as many parents of young children faced increased parenting stress during the COVID-19 pandemic and beyond [16,17,37,45], since the crisis mode persisted with the subsequent war in Ukraine and rising inflation in Germany. In fact, the needs of young families were already not being fully met during the pandemic as ECI usage rates decreased by 28% as a result of pandemic-related restriction measures [40]. By increasing barrier-free accessibility and provision through digitally delivered evidence-based interventions, more parents with young children may be reached [46,47]. With widespread distribution and use of mobile devices there is also high potential for reaching low-income target groups, as shown in the evaluation of the ‘Afinidata’ digital parenting intervention project in remote parts of Latin America [48]. In Germany, around 96% of 20–39 year olds owned a smartphone/mobile phone in 2021 [49]. Digitization was therefore also a common way to maintain support during times of restrictions in the COVID-19 pandemic [13] and has great potential beyond extending existing ECI services and contributing to quality development [50]. In addition, digital tools can help to diversify ECI communication channels, as was the case in this study, to create even more touchpoints alongside the existing ones (e.g., pediatricians, local networks as the KoKis). Further, within such digital applications parents could be directly connected with existing face to face support services (e.g., via appointment links, telephone-buttons). Currently, most of the parents in the CoronabaBY study stated that they initiated the contact with the service themselves (nearly 90%).

Fortunately, many of the families of the presented RCT study were already aware of ECI services. Those parents in need of support and searching for help had no (36.1%) or only a few days (47.8%) waiting time until access and mostly rated the services as helpful or very helpful (34.3% and 30.4%).

The CoronabaBY study shows several strengths and limitations (see also [16,17,37]. It is the first pandemic study in Germany on families with young children up to 3 years on their experience of psychosocial stresses and the awareness and usage of ECI services. Furthermore, the study covers a large pandemic period and sample, uses a validated instrument (parenting stress index, PSI) and shows a very high level of data completeness.

Looking at the limitations, there could be questions about generalizability as the participants were mostly well-off and more highly educated German mothers living in Bavaria. Although this is often common in social studies [51], the impact of our study might be underestimated, since the participants were probably already well-informed. Moreover, only app users within participating pediatric practices could be recruited which may have led to a selection bias as online surveys are known to be often self-selective (e.g., higher participation rates among people with a higher level of education) [52,53] and families without digital access or those not using the app could not be included. This in turn can lead to sampling bias and also affect the generalizability of the data. However, the widespread use of mobile devices in Germany is likely to provide a large part of the target group with general access to digital services.

Furthermore, conclusions regarding the awareness and usage of ECI services and the effect of the intervention should be drawn with caution given the relatively low access rate to the app module in the IG. Whether the low rate was due to the design and content of the module itself, to the placement within the app or to other factors (e.g., lack of interest in the topic, existing knowledge of ECI services) cannot be conclusively assessed. Nevertheless, for those in the IG who accessed the module, a positive effect on the usage of ECI services could be shown for the post-test.

It should also be mentioned that the questions on awareness and usage of ECI services were created by the study team due to a lack of instruments. They have not been structurally validated. However, the design with answer options at the nominal data level is comparable to similar queries on ECI services in the German study KiD 0–3 [40]. Since the measurement of awareness and usage of ECI services was only possible through self-reporting and partly retrospective questions (e.g., on previous usage), socially desirable answers and recall biases cannot be ruled out.

## 5. Conclusions

Families with very young children should be addressed at an early stage with appropriate information about and access to Early Childhood Intervention services (ECI). Digital applications have the potential to increase awareness as well as usage by making ECI services more visible, diversifying communication channels and acting as an intermediary to access professional advice. Although no definitive statement can be made about the effect of the app-based intervention of the present RCT on awareness of ECI, a significant increase in awareness was observed in the entire study population. The usage rates of ECI services, however, were positively influenced by the intervention and thus significantly higher in the intervention group after accessing the module, especially among highly stressed parents with increased need of support. The focus should therefore continue to be on the further development and evaluation of such digital strategies, e.g., in the form of prominent positioning of ECI on the pages of the app and a direct connection to existing face-to-face support services via appointment links or telephone buttons. Digital approaches also promise potential for reaching more disadvantaged target groups (e.g., low-income) due to a widespread distribution and use of mobile devices in the total population of Germany.

## Figures and Tables

**Figure 1 healthcare-13-02000-f001:**
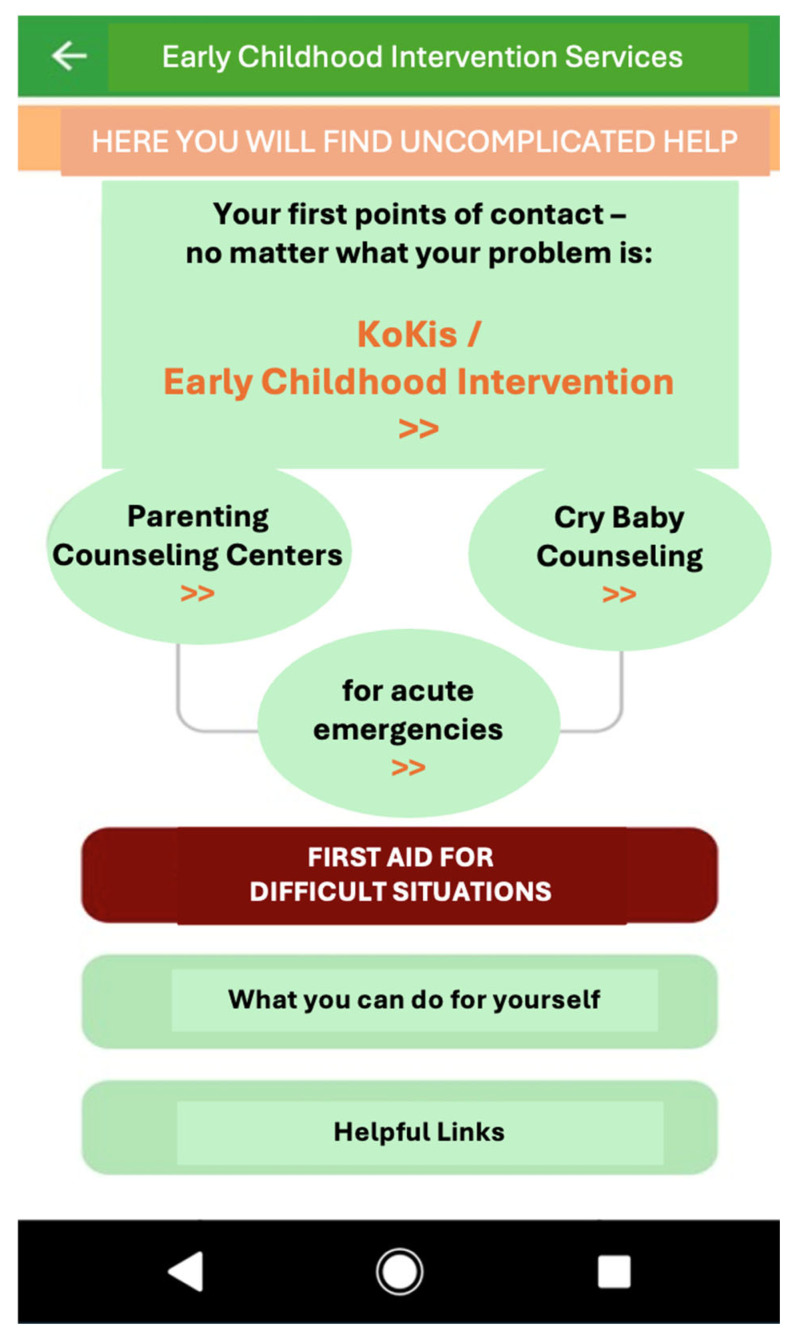
App-based intervention: screenshot of the navigation page of the information module (translated into English, originally in German).

**Figure 2 healthcare-13-02000-f002:**
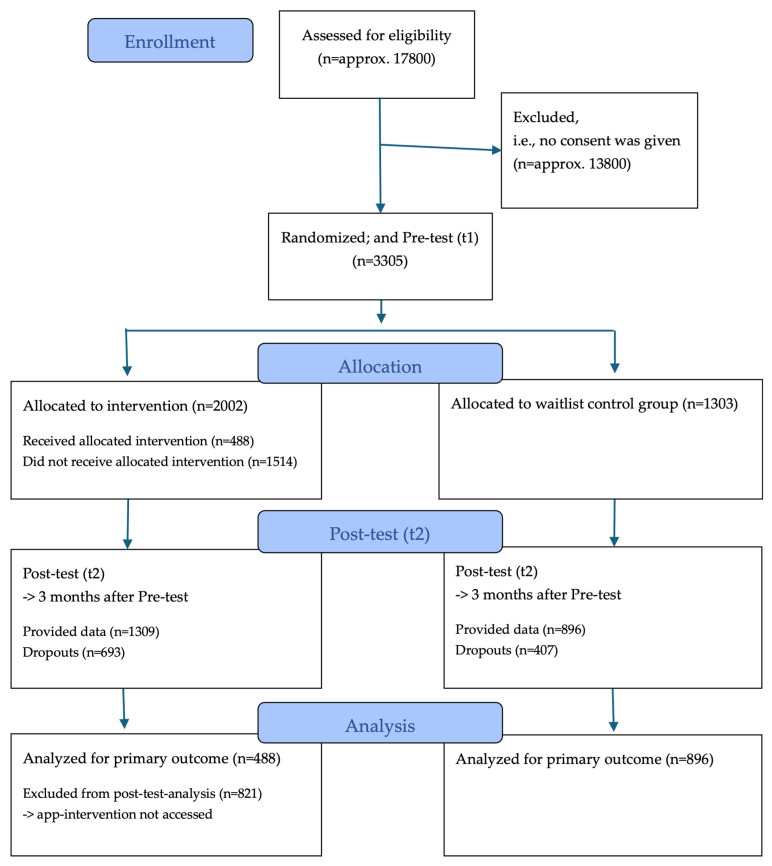
CONSORT participant flow diagram.

**Figure 3 healthcare-13-02000-f003:**
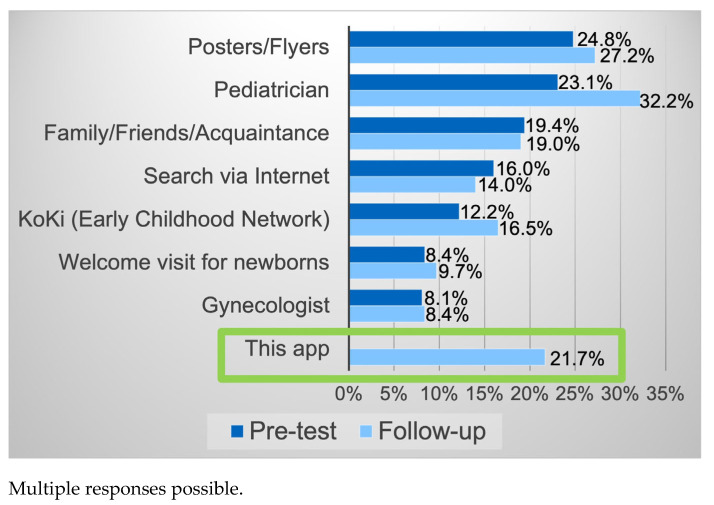
Source of knowledge of Early Childhood Intervention services.

**Table 1 healthcare-13-02000-t001:** Sample characteristics.

Sociodemographics of Parents	Total Sample n (%)	Intervention Group n (%)	Waitlist Control Group n (%)	*p* ^b^
Mothers	3072 (92.9)	1857 (92.7)	1215 (93.2)	0.579
Mother tongue German	3053 (92.3)	1846 (92.2)	1207 (92.6)	0.619
Educational level (at least high school diploma)	1965 (59.5)	1210 (60.4)	755 (58)	0.121
Comfortable financial situation ^a^ (self-assessment)	1762 (53.3)	1099 (54.9)	663 (50.9)	0.006
**Age of parents and child**	M (SD)	M (SD)	M (SD)	
*M*_age_ participants in years	33.7 (5.0)	33.7 (5.1)	33.7 (5.0)	0.826
*M*_age_ child in months	16.7 (12.1)	16.4 (12.1)	17.2 (12.1)	0.064
**Parenting Stress** **(acc. to parenting stress index)**				
No findings	1817 (55.0)	1104 (55.1)	713 (54.7)	
Stressed	1117 (33.8)	670 (33.5)	447 (34.3)	0.856
Highly stressed	371 (11.2)	228 (11.4)	143 (11.0)	

^a^ Outcome (scale) dichotomized into: high/comfortable (very large/large additional purchases possible) vs. low (small/very small/no additional purchases possible). ^b^ Intervention group vs. waitlist control group.

**Table 2 healthcare-13-02000-t002:** Awareness and usage of Early Childhood Intervention services (ECI) for study group and over time.

Outcome Variable	Intervention Group (IG) n (%)	Waitlist Control Group (WCG) n (%)	*p*	Cramer’s V
**Awareness of ECI**				
t1 (pre-test)	926 (70.6)	598 (73.2)	0.193	0.028
t2 (post-test)	271 (86.9) ^a^	472 (85.7)	0.625	0.017
Follow-up	173 (90.6) ^a^	273 (86.1)	0.137	0.066
**Usage of ECI**				
t1 (pre-test)	691 (34.5)	486 (37.3)	0.100	0.029
t2 (post-test)	59 (12.1) ^a,b^	107 (8.2) ^b^	0.012	0.060
Follow-up	28 (9.9) ^a,c^	51 (9.8) ^c^	0.961	0.002

^a^ at least app module accessed, ^b^ further usage since pre-test (t1), ^c^ further usage since post-test (t2); sign. differences in awareness of ECI from t1 (pre-test) to t2 (post-test) in IG and WCG (*p* < 0.001).

**Table 3 healthcare-13-02000-t003:** Awareness and usage of Early Childhood Intervention services (ECI) according to parenting stress levels (PSI).

Outcome Variable	Not Stressed n (%)	Stressed n (%)	Highly Stressed n (%)	*p* (*V*) (Not Stressed vs. Stressed)	*p* (*V*) (Stressed vs. Highly Stressed)	*p* (*V*) (Not Stressed vs. Highly Stressed)
**Awareness** **of ECI**						
t1 (pre-test)	958 (76.6)	465 (68.1)	101 (51.5)	<0.001 (0.093)	<0.001 (0.144)	<0.001 (0.194)
Follow-up	275 (89.9)	123 (87.9)	48 (77.4)	0.525 (0.030)	0.058 (0.134)	0.006 (0.142)
**Usage of ECI**						
t1 (pre-test)	567 (31.2)	434 (38.9)	175 (47.2)	<0.001(0.078)	0.005 (0.073)	<0.001 (0.127)
Follow-up	49 (7.0) ^a^	46 (11.2) ^a^	31 (17.4) ^a^	0.013 (0.074)	0.042 (0.084)	<0.001 (0.146)

^a^ further usage of ECI since last survey stage; *V* = Cramer’s V; significant differences in awareness of ECI from t1 (pre-test) to follow-up in all PSI subgroups (*p* < 0.001).

## Data Availability

The datasets presented in this study are not readily available because there is no consent from the participants for sharing. Request for access to the datasets should be directed to catherine.buechel@tum.de.

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
