# Peer review of "Effects of an App-Based Intervention to Improve Awareness and Usage of Early Childhood Intervention Services During the COVID-19 Pandemic: Randomized Controlled Trial of the CoronabaBY Study from Germany"

_healthcare, 2025, doi:10.3390/healthcare13162000_

Round 1
Reviewer 1 Report
Comments and Suggestions for Authors
This study is significant and precautionary in promoting access to Early Childhood Intervention services. A large sample size and a well-organized randomized controlled trial design were used.
Comments:
This study is significant and precautionary in promoting access to Early Childhood Intervention services. A large sample size and a well-organized randomized controlled trial design were used. However, important points to consider are as follows:
Abstract
- The abstract is generally well structured, but no mention of effect sizes or statistical significance levels. Significant findings should be included in the summary along with p-values
Introduction
- The introduction is extremely long and contains repetitive explanations. It could be simplified to maintain the reader's interest.
Method
- The inclusion and exclusion criteria of the study were not clearly stated. For example, were parents with psychiatric diagnoses included in the study? Were those already actively using ECI services excluded?
- The application module and its contents are briefly described; only general headings are included. Essential details such as the contents' scope, format, presentation language, or interaction level are not provided. It would be helpful to present it as supplementary material.
- The measurements evaluated under "Awareness and Usage of Early Childhood Intervention (ECI)" seem to have been made with study-specific questions instead of a known, valid, and reliable scale. The questions used for the measurement (e.g., "Do you know that there are free, non-binding support services for young families…?") were presented as descriptive questions only, without conducting a structural validity analysis. If the authors developed this measurement tool, they should provide information about criterion validity and reliability. Alternatively, previously validated measurement tools (if any) should be used, or at least such validity limitations should be clearly stated in the discussion section. In addition, question examples should be provided as supplementary material (supplementary file).
- Parenting Stress Index should include what high or low scores indicate. (For example, a high score indicates high stress.)
Results
- In the sample characteristics presented in Table 1, comparisons of the intervention and control groups are given, but only the p-value for one variable (financial status) is indicated (with an *). All other variables should be statistically tested and reported for group differences.
Discussion
- The discussion is generally balanced and consistent with the results. It is clearly stated that the intervention only affected usage and that there was no significant difference in awareness. However, the details of the application and the low access rate of the application (37.3%) should have been emphasized more as a critical limitation.
Author Response
Response to Reviewer 1 Comments |
||
1. Summary |
|
|
Thank you very much for taking the time to review this manuscript. Please find the detailed responses below and the corresponding revisions/corrections in track changes in the re-submitted files (the line numbers given in the responses refer to the mode in Word with all markups displayed). |
||
2. Questions for General Evaluation |
Reviewer’s Evaluation |
Response and Revisions |
Does the introduction provide sufficient background and include all relevant references? |
Yes |
|
Is the research design appropriate? |
Can be improved |
Responses are given in the point-by-point response letter below. |
Are the methods adequately described? |
Can be improved |
Responses are given in the point-by-point response letter below. |
Are the results clearly presented? |
Yes |
|
Are the conclusions supported by the results? Are all figures and tables clear and well-presented? |
Yes
Yes |
|
3. Point-by-point response to Comments and Suggestions for Authors |
||
|
||
Response 1: Thank you for pointing this out. We agree with this comment. Therefore, we have added effect sizes resp. p-values in the abstract (see page 1, line 29 and line 31). |
||
Introduction Comments 2: The introduction is extremely long and contains repetitive explanations. It could be simplified to maintain the reader's interest. |
||
Response 2: We agree. We have, accordingly, shortened the introduction by cutting out repetitions and less relevant details (see deletions in track changes on page 2), and modified some sentences/one section (see page 2, line 68-73). |
||
Method Comments 3: The inclusion and exclusion criteria of the study were not clearly stated. For example, were parents with psychiatric diagnoses included in the study? Were those already actively using ECI services excluded? Response 3: Thank you. To make this point clearer, we have explained/specified the possibilities of applying inclusion/exclusion criteria more precisely (see page 4, line 265-267). Furthermore, we decided to reanalyze the primary outcome `Awareness´ by excluding those participants who stated in the pre-test that they had already used ECI services before, to counteract memory effects and achieve more distinct results. We added/corrected this in the results (see page 9, line 4210-425 and table 2 on page 10, line 465-468), in the discussion (page 12, line 712-717) and in the statistics section (see page 6, line 361-362). Regarding the primary outcome `Usage´, we did not apply this filter because for this outcome we also wanted to include those participants who used ECI services again, not only for the first time (see page 6, line 363-364). Comment 4: The application module and its contents are briefly described; only general headings are included. Essential details such as the contents' scope, format, presentation language, or interaction level are not provided. It would be helpful to present it as supplementary material. Response 4: Thank you for this advice. We added more of the above-mentioned details in the text about the module (see page 4, line 282-295) and created supplementary material with more explanatory screenshots of the app module (see supplementary material 1). Comment 5: The measurements evaluated under "Awareness and Usage of Early Childhood Intervention (ECI)" seem to have been made with study-specific questions instead of a known, valid, and reliable scale. The questions used for the measurement (e.g., "Do you know that there are free, non-binding support services for young families…?") were presented as descriptive questions only, without conducting a structural validity analysis. If the authors developed this measurement tool, they should provide information about criterion validity and reliability. Alternatively, previously validated measurement tools (if any) should be used, or at least such validity limitations should be clearly stated in the discussion section. In addition, question examples should be provided as supplementary material (supplementary file). Response 5: Thank you for this comment. We would like to explain our approach in more detail. Since there are no existing validated questionnaires to assess knowledge and usage of Early Childhood Intervention services (ECI), we had to design them ourselves. Since these questions did not ask about constructs or symptom complexes (as is the case, for example, with the assessment of Parenting Stress), these questions were created by the study team and largely kept simple at a nominal data level. They have not been specifically validated, but their design/answer options are comparable to the questions resp. answer options concerning ECI services in the German comparative study KiD 0-3. To address possible validity limitations, we have added a statement in the discussion section (see page 14, line 835-838). We have also created a supplementary file with the questions created and used (see supplementary material 2). Comment 6: Parenting Stress Index should include what high or low scores indicate. (For example, a high score indicates high stress.) Response 6: Thank you for this advice, we have added a sentence explaining what a high score indicates (see page 6, line 336-337).
Results Comment 7: In the sample characteristics presented in Table 1, comparisons of the intervention and control groups are given, but only the p-value for one variable (financial status) is indicated (with an *). All other variables should be statistically tested and reported for group differences. Response 7: Thank you for pointing this out. We agree with this comment. Therefore, we have added a column to Table 1 with p-values for all variables regarding group differences (see table 1, page 9, line 403-405). Financial status was the only variable with a significant difference between the two groups. Since ECI services are low-threshold, cost-free and available to everyone (and also the intervention module was cost-free), we did not actively control for financial status in the analyses.
Discussion Comment 8: The discussion is generally balanced and consistent with the results. It is clearly stated that the intervention only affected usage and that there was no significant difference in awareness. However, the details of the application and the low access rate of the application (37.3%) should have been emphasized more as a critical limitation. Response 8: Thank you for this remark. We have accordingly addressed these points more detailed in the discussion (see page 13, line 760-770 and page 14, line 828-834). |

Reviewer 2 Report
Comments and Suggestions for Authors
Many thanks for the opportunity to review this manuscript. Several aspects of the study design and interpretation require clarification.
1. Justify the exclusion of non-users of the intervention module from post-test and follow-up analyses, and address the implications for internal and external validity.
2. Reassess the strength of causal inferences regarding intervention effects on service utilization, given the lack of significant group differences in awareness and the potential for measurement-induced changes across groups.
3. Acknowledge and discuss the likelihood of selection bias arising from app-based recruitment, particularly concerning socioeconomic status and digital access.
4. Align interpretation of results more closely with the observed intervention uptake, particularly in relation to the limited engagement with the app module and comparable awareness increases across groups.
5. Specify whether baseline differences between groups (e.g., financial status) were statistically controlled in outcome analyses, and clarify their potential influence on results.
Additional comments:
- The exclusion of intervention group participants who did not access the app module from post-test and follow-up analyses introduces potential attrition bias and limits the external validity of findings. Was this a per-protocol analysis? Please justify this decision and consider also reporting intention-to-treat (ITT) results.
- The intervention effect is measured via self-reported awareness and usage, which are subject to social desirability and recall biases.
- The recruitment was limited to app users within pediatric practices. This exclusion of families without digital access or those not using the app raises concerns about sampling bias and generalizability.
Author Response
Response to Reviewer 2 Comments |
||
1. Summary |
|
|
Thank you very much for taking the time to review this manuscript. Please find the detailed responses below and the corresponding revisions/corrections in track changes in the re-submitted files (the line numbers given in the responses refer to the mode in Word with all markups displayed). |
||
2. Questions for General Evaluation |
Reviewer’s Evaluation |
Response and Revisions |
Does the introduction provide sufficient background and include all relevant references? |
Yes |
|
Is the research design appropriate? |
Yes |
|
Are the methods adequately described? |
Can be improved |
Responses are given in the point-by-point response letter below. |
Are the results clearly presented? |
Yes |
|
Are the conclusions supported by the results? Are all figures and tables clear and well-presented? |
Can be improved
Yes |
Responses are given in the point-by-point response letter below. |
3. Point-by-point response to Comments and Suggestions for Authors |
||
Comments 1: Justify the exclusion of non-users of the intervention module from post-test and follow-up analyses, and address the implications for internal and external validity. Response 1: Thank you for pointing this out. We decided to include only those participants of the intervention group in the RCT who accessed the intervention (i.e., viewed/used the app info module on ECI services) to be able to attribute effects at post-test and follow-up to the usage of the intervention (compared to the waitlist control group). With regard to external validity, these results can of course only be generalized to the group of module users within the intervention group (IG). In order to complete the results and draw more precise conclusions, we have taken up your suggestion and additionally conducted an ITT analysis (see comment and response 6).
Comments 2: Reassess the strength of causal inferences regarding intervention effects on service utilization, given the lack of significant group differences in awareness and the potential for measurement-induced changes across groups. Response 2: Thank you. We tried to formulate and interpret the postulated intervention effects more moderately (see page 12, line 745-750, page 13, line 760-770 and page 14, line 828-834). At the same time, we saw a significant difference between those in the intervention group who used the app module and the waitlist control group with respect to usage rates at post-test. The derived assumption of an effect is supported by the fact that this difference between the groups is no longer evident when all participants of the intervention group (including those who did not access the app module) are included in an intention-to-treat analysis (see also ITT-analysis, comment and response 6). Of course, no causal conclusions are possible. We assume that module usage contributed along with other factors (e.g., measurement-induced changes) for which we could not control. We added more detailed explanation (see page 12, line 745-750, page 13, line 760-770 and page 14, line 828-834).
Comments 3: Acknowledge and discuss the likelihood of selection bias arising from app-based recruitment, particularly concerning socioeconomic status and digital access. Response 3: We agree with this point and have therefore discussed the potential impacts of an app-based recruitment in more detail with regard to selection bias and the exclusion of families without digital access or not using the app (see page 13, line 806-810 and page 14, line 825-827).
Comments 4: Align interpretation of results more closely with the observed intervention uptake, particularly in relation to the limited engagement with the app module and comparable awareness increases across groups. Response 4: Thank you for pointing this out. We agree and have therefore added more detailed explanations (see page 12, line 745-750, page 13, line 760-770 and page 14, line 828-834). For further details see also comment/response 2. We are aware that the postulated effect can only be transferred to the group of parents within the IG who actually access(ed) the intervention module, especially in view of the generally limited engagement with the app module among all IG-participants.
Comments 5: Specify whether baseline differences between groups (e.g., financial status) were statistically controlled in outcome analyses, and clarify their potential influence on results. Response 5: Thank you for this comment. Financial status was, in fact, the only variable with a significant difference between the two groups at baseline. Because ECI services are low-threshold, cost-free, preventive and available to everyone (and the app intervention module was also cost-free), we expected no potential influence on the results and therefore did not actively control for financial status in the analyses of awareness and usage of ECI services. Educational status, however, could have had an influence, e.g., regarding potential higher interest/awareness of the research topic among parents with higher education. However, there was no significant difference in educational status between the groups at baseline. Therefore, we did not control for this variable either.
Additional Comments:
Comments 6: The exclusion of intervention group participants who did not access the app module from post-test and follow-up analyses introduces potential attrition bias and limits the external validity of findings. Was this a per-protocol analysis? Please justify this decision and consider also reporting intention-to-treat (ITT) results. Response 6: Thank you for pointing this out. We agree that excluding a part of the participants from the intervention group may lead to attrition bias and a lack of external validity. Nevertheless, we decided on this selection to be able to attribute observed differences/effects more precisely to module usage, especially given that only about one-third of the IG-parents accessed the intervention module.
Comments 7: The intervention effect is measured via self-reported awareness and usage, which are subject to social desirability and recall biases. Response 7: Thank you for this advice. Unfortunately, the measurement of awareness and usage of ECI services was only possible via self-report and partly retrospective questions (e.g., regarding previous usage). We are open to suggestions for future assessment of these outcomes. To address the possibilities of socially desirable answers as well as recall biases, we added these aspects in the discussion (limitations) (see page 14, line 835-841).
Comments 8: The recruitment was limited to app users within pediatric practices. This exclusion of families without digital access or those not using the app raises concerns about sampling bias and generalizability. Response 8: We agree with this comment. Therefore, we have added more details on these aspects (see page 13, line 806-810 and page 14, line 825-827). Because of the widespread distribution of mobile devices in our target group, we assume a high degree of digital accessibility (see page 14, line 825-827). Overall, we would assume that users of the app represent the population to a certain extent. However, we do not know whether results are transferable to non-app users. |

Round 2
Reviewer 1 Report
Comments and Suggestions for Authors
I see that all suggested changes have been made meticulously. The article is suitable for publication in its current form.